# Extending the model of children's conduct problems: A cross-sectional study of the interaction of maternal temperament and character, maternal parenting practices, and their child's effortful control

**Maor Yeshua**[1]*, **Claude Robert Cloninger**[2], **Ada H. Zohar**[3,4,5], **Andrea Berger**[1,6]

**1** Department of Psychology, Ben-Gurion University of the Negev, Beer Sheva, Israel, **2** Department of Psychiatry, Washington University, St. Louis, Missouri, United States of America, **3** Department of Behavioral Sciences, Ruppin Academic Center, Emek Hefer, Israel, **4** Graduate Program in Clinical Psychology, Ruppin Academic Center, Emek Hefer, Israel, **5** Graduate Program in Gerontological Clinical Psychology, Ruppin Academic Center, Emek Hefer, Israel, **6** School of Brain Sciences and Cognition, Ben-Gurion University of the Negev, Beer Sheva, Israel

* maor.yeshua@gmail.com

## Abstract

### Introduction

Children's externalizing problems in kindergarten are risk-factors that can explain psychopathology at adolescence and adulthood. Hence, it is important to study the complex and multiple-layer processes that might explain and reduce their occurrence. Among the most important moderating factors are parental caregiving practices, especially maternal sensitivity, which may depend on the parent's temperament and character. We hypothesized that maternal effortful control (a temperament trait) and cooperativeness (a character trait) interact to facilitate maternal sensitivity and their children's self-regulation and conduct.

### Method

One hundred and sixty-three mothers and their children participated in a cross-sectional study. The mothers completed self-report measures of effortful control (ATQ), cooperativeness (TCI), expressive encouragement (CCNES), and their children's effortful control (CBQ) and conduct problems (SDQ). They also performed the Etch-A-Sketch task with their children, which was coded offline for behavioral indicators of mothers' cooperativeness. A composite factor of cooperativeness was derived from the behavioral and self-report measures. Hayes's mediation cascade model (Model 6) was used to test our primary hypothesis.

**Data availability statement:** Availability of data and material: All data and material are available online (https://github.com/MaorYeshua/EC-x-CO-ms.git). Code availability: All code is available online (https://github.com/MaorYeshua/EC-x-CO-ms.git).

**Funding:** This study was partially supported by the Israel Scientific Foundation (ISF; Grant number 533/20).

**Competing interests:** The authors have declared that no competing interests exist.

## Results

The model explained 22.5% of the child's conduct problems. Within the mediation model, the mother's cooperativeness moderated the relation between her effortful control and expressive encouragement of her child's emotions ($\beta = .16$, $t(152) = 2.62$, $p = .010$). Her expressive encouragement contributed to the child's effortful control ($\beta = .27$, $t(151) = 2.55$, $p = .012$) and fewer conduct problems ($\beta = -.21$, $t(150) = -2.67$, $p = .008$). The full mediation path was found to be significant (B $= -.01$, $95\%$ Bootstrap CI $[-.025, -.001]$).

## Discussion

The findings suggest that maternal temperament and character trait interact in their effects on the mothers' behaviors, and, through them, on their children's behaviors.

## Introduction

Conduct problems in childhood are characterized by frequent temper tantrums, disobedience, fighting with other children, lying and stealing [1]. The earlier the age of the manifestation of conduct problems, the greater is the risk of chronic misconduct [2]. Moreover, Reef et al. [3] in their 24-year follow-up study showed that the presence of externalizing problems and aggressive behavior during childhood and adolescence predicted the presence of such behaviors 24 years later. Thus, it is important to investigate the complex and multiple-layer process resulting in a child's conduct problems early in life, as in kindergarten-aged children.

Of the factors that contribute to a child's behavior, the key predictors are their own effortful control (EC; the temperamental construct of self-regulation [4–7]) and the mother's EC and parenting practices [8–10]. Theoretically and empirically, EC measures an individual's executive attention and ability to inhibit impulsive conduct and emotional outbursts [7,11,12]. Mothers with better EC tend to engage in more supportive and sensitive parenting practices than others [8]. For example, these mothers are more sensitive and involved [13,14] and establish their expectations for their children more clearly [15]. Such maternal practices, in turn, promote better EC in the child [10,16,17], which is expressed in behavior as better adaptive functioning, particularly fewer conduct problems [9,18–20].

Recently, another aspect was added to this mediation cascade, as a possible predictor of the mothers' parenting – the mothers' character traits [21]. In their systematic review, Yeshua and Berger [21] discussed the complex construct of maternal personality that includes temperamental traits (e.g., EC) and character traits that develop later in life (e.g., work ethic or respect for others). They further suggested that interactions between these traits may help explain variations in maternal parenting behavior. Although there are some findings that support this notion (e.g., [22,23]), this idea has not been thoroughly investigated [21]. The current study focused on cooperativeness (CO). CO is a character trait that is defined as being socially tolerant

and tender-hearted, empathic, helpful, and compassionate. Its full development can extend to unconditional acceptance of others along with the willingness to help others unselfishly [24]. As suggested in the review by Yeshua and Berger [21], it is possible that the relationship between mothers' EC and their sensitive parenting is being moderated by their CO. Theoretically, mothers with good inhibitory control and executive attention (i.e., stronger EC) might be more efficient and organized, but it does not follow that they would also be empathic and tolerant towards their child in times of distress (i.e., higher levels of CO). Hence, this study aimed to examine the plausible interaction between the mothers' temperament (in the form of EC) and their character (in the form of CO), for explaining their caregiving practices, as well as their children's EC and conduct problems.

Mothers' CO could moderate the effect of EC, especially when explaining mothers' sensitive caregiving practice of expressive encouragement, which is when the mothers actively encourage their children's expression of negative emotions [25]. In order to use expressive encouragement when their children become angry or upset, the mothers need both high EC to moderate their own distress in the presence of their children's distress, and high CO to be empathic, tolerant and compassionate. In other words, they need to be able to identify their children's distress and be able to let them express it without being maladaptively affected by it. There is some empirical support for this theoretical notion. Scott and Hakim-Larson [26] showed that mothers who had better EC and lower negative affectivity, as well as their children, showed lower rejection and disapproval of the children's emotions. Moreover, these findings correspond well with Edler and Valentino's [27] systematic review that offered that parents' EC is associated with better practice of discussions of emotions with children; however, as they have stated, surprisingly, the literature on the matter is scarce and further research is needed. In more general terms, better EC of the mother was found to explain more adaptive parenting practices [9,10,28]. Alongside these findings, it was found that mothers who have high levels of CO seem to be more inclined to exert authoritative and caring parenting [29,30]. They also use better parenting practices in general when dealing with their child's negative emotions [9]. However, as suggested above, the relationship between EC and CO, specifically when explaining expressive encouragement, might be more complex.

Following the mediation cascade described earlier, better emotion socialization, and specifically expressive encouragement, explained better emotion regulation [26], less emotional lability and less conduct problems [31]. As broadly presented earlier, the expressive encouragement of the child also corresponds with the more general concept of maternal sensitivity. Many empirical findings support the relation between better maternal sensitivity and better self-regulation and behavior of the child (see systematic review by [32], and meta-analyses by [33,34]). Hence, the relations between maternal expressive encouragement and children's EC and conduct problems were expected to be related in a similar manner.

To conclude, no previous study has explored the interaction between mothers' EC and CO and its role in the intergenerational transmission of self-regulation and the behavioral outcome of children. Therefore, the current study explored the way that the interaction between mothers' EC and CO explains their parenting practices, which in turn would explain their children's EC and conduct problems.

## Method

### Participants

A sample of 175 mothers of neuro-typical kindergarten-aged children was recruited via online platforms (such as Facebook and Instagram) and they were asked to answer a survey and then were invited to a two-hour lab visit. During this visit, the mother and their child performed a defined set of tasks. After data collection and scoring of the relevant behavioral data, 12 dyads were omitted from the main analysis due to missing values; hence, the final sample in the main analysis was 163 dyads.

The mothers' ages ranged between 28–51 years ($M = 36.69 \pm 4.45$) and years of education ranged between 7–30 years ($M = 17.62 \pm 3.16$), which when compared to national data, indicates the mothers in the current sample were slightly more educated [35]. Their household income was diverse with half having higher/lower income compared to the monthly

demographic average. Their children's sex was evenly distributed (53% males; $n = 92$) and their mean age was 4.55 years ± .47 (3.00–6.46).

## Ethical standards and procedure

The study received ethical approval from the Human Subjects Research Committee of Ben-Gurion University of the Negev (protocol no. 1985−2) and the Helsinki Committee (protocol no. 0342-15-SOR). Data collection was conducted between November 23, 2021, and September 5, 2023.

Participants were recruited via online platforms and provided oral consent during an initial screening call, followed by the completion of an online informed consent form. Upon arrival at the laboratory, they also signed the Helsinki consent form concerning both their own and their child's participation. Children provided oral assent at the time of their visit. To ensure confidentiality, each participant was assigned a unique identification code used throughout all phases of the study. Upon completion of participation, individuals received compensation as agreed upon with the principal investigator.

## Procedures and measures

Participants answered a survey built on the Qualtrics platform that included: (1) the Adult Temperament Questionnaire (ATQ–Short Form; [11]), (2) the Coping with Children Negative Emotions Scale (CCNES; [25]), (3) the Agreeableness and Conscientiousness sub-scales from the Big-Five Inventory (BFI; [36]), (4) the Cooperativeness and Self-Directedness sub-scales from the Temperament and Character Inventory – Revised (TCI-R; [37]), (5) the Children Behavior Question-naire (CBQ; [38]), (6) the Strength and Deficits Questionnaire (SDQ; [1]), and (7) the Confusion, Hubbub, and Order Scale (CHAOS; [39]).

Dyads that came to the lab performed several tasks, including EEG recording, a computerized Emotional Day-Night task, the Raven task, and two filmed tasks (the Transparent Box and the Etch-A-Sketch). The hypotheses of the present study were tested based on the ATQ, CBQ, CCNES, TCI, CHAOS and SDQ questionnaires, as described in the following sections, as well as on behavioral coding of the mother's behavior drawn from the Etch-A-Sketch task (see Table 1 for summarized measures). The authors had permission to use these instruments from the copyright holders.

**Mothers' and children's effortful control.** Assessment was done using a sub-scale from the Adult Temperament Questionnaire (ATQ–Short Form; [11]) and the Children's Behavior Questionnaire (CBQ; [7]). The EC of the mothers was computed as the mean value of 18 items (Cronbach's α = .76). The EC of the children was computed as the mean value of 26 items (Cronbach's α = .84).

Table 1. Summarized measures.

| Theoretical variable | Operationalization | Variable role in the mediation model |
|---|---|---|
| Mothers' effortful control | • Average value based on self-reports items using the Adults Temperament Questionnaire | Independent variable + interaction with mothers' cooperativeness |
| Mothers' cooperativeness | • Average value based on self-reports using the Temperament and Character Inventory – Revised<br>• Four average ratings of behavioral codes that were coded in the Etch-A-Sketch task<br>• A factor was extracted using these five scales | Independent variable + interaction with mothers' effortful control |
| Mothers' expressive encouragement | • Average value based on self-reports using the Coping with Children's Negative Emotions Scale | First mediator |
| Children's effortful control | • Average value based on mothers' reports using the Children's Behavior Questionnaire | Second mediator |
| Children's conduct problems | • Average value based on mothers' reports using the Strengths and Difficulties Questionnaire | Dependent variable |

**Mothers' character – self-reported cooperativeness.** Assessment was done using a sub-scale from the Temperament and Character Inventory – Revised (TCI-R; [37]). The cooperativeness (CO) of the mothers was computed as the mean value of 20 items (Cronbach's α = .80).

**Etch-A-Sketch – cooperativeness codes.** A 7-minute mother-child interaction [40] was carried out using an Etch-A-Sketch. This task involves a degree of cooperation, responsiveness and compliance between mothers and children. The Etch-A-Sketch toy has two dials – one for moving the drawing point vertically and the other for moving it horizontally. The researcher assigned each individual one dial to use; both of the participants were instructed not to touch the other's dial. Each 7-minute recording was divided to four segments of 1 minute and 45 seconds. Two coders coded the interaction using an adapted version of the Parent-Child Interaction System (PARCHISY; [41]). Based on a 10-recording sample, the ratings were found reliable (Cohen's Kappa = .74; ICC = .93). Each mother in each coding had four ratings, which were summed and divided by four in order to calculate a proportional mean value. In this study, we integrated four additional theory-driven codes into the coding, based on two sub-scales of CO, according to Table 2 in Cloninger et al.'s [42] paper. The first theoretical sub-scale that was translated into two codes was social acceptance vs. intolerance, which was designed as a bipolar scale. The second theoretical sub-scale that was translated into two codes was empathy vs. social disinterest. The two poles of each scale can be expressed by the same person in different situations at different times. Therefore, in practice, especially in an ecological setting, both types of behavioral manifestations can occur in different moments of the interaction between the mother and the child; hence, they can be coded as distinct and not mutually exclusive scales. See full description in S1 Appendix.

**Parental practices.** To measure parental practices, the Coping with Children's Negative Emotions Scale (CCNES; [25]) was used (72 statements). Adaptive parenting was measured using the following sub-scales: problem-focused responses (PF; helping the child solve the problem that caused the distress, α = .84), emotion-focused responses (EF; helping the child feel better, α = .84), and expressive encouragement responses (EE; actively encouraging children's expression of negative emotions, α = .90). Aspects of maladaptive parenting were measured as follows: minimization responses (MR; discounting or devaluing the child's negative emotions/problem, α = .81), punitive responses (PR; using verbal or physical punishment to control the expression of negative emotion, α = .82), and distress responses (DR; becoming adversely aroused/distressed by a child's negative emotion, α = .72). Observed variables were the mean value of each sub-scale, where higher values indicate higher levels of reported constructs.

**Children's conduct problems.** Assessment was done using a sub-scale from the Strengths and Difficulties Questionnaire (SDQ; [1]) as the mean of five items (Cronbach's α = .64) regarding disobedience, temper, tendencies to fight, steal, and lie.

**Background variables.** As socioeconomic status is a risk factor for the development of psychiatric disorders [43], we examined it as a possible covariate. A factor was extracted using the Principal Component Analysis method loaded by mothers' years of education (.56), household income (.80), number of cars (.69) and number of household rooms

**Table 2. Cooperativeness variables zero-order correlations.**

| Variables | 1. | 2. | 3. | 4. |
|---|---|---|---|---|
| 1. CO | – | | | |
| 2. Social acceptance | .20** | – | | |
| 3. Intolerance | −.18* | −.18* | – | |
| 4. Empathy | .05 | .29*** | −.09 | – |
| 5. Social disinterest | .09 | .05 | .36*** | .20** |

*N* = 172. Higher values point to more extreme responses in the expected direction. CO was self-reported using the Temperament and Character Inventory (TCI) and all other variables are observation-based scores in the Etch-A-Sketch task. CO = cooperativeness. *p < .05, **p < .01, ***p < .001, two-tailed.

(.69). Moreover, as household chaos and intelligence (of mother and child alike) are related to one another and have an effect on the child's outcomes [44,45], they were also considered as possible covariates and were assessed using the Confusion, Hubbub, and Order Scale (CHAOS; [39]), which is the mean value of 15 items (Cronbach's α = .83). In addition, the Raven Standard Progressive Matrices (SPM) and Colored Progressive Matrices (CPM) tests (adult and children Raven tests, respectively; [46]) were used. The Raven tests include three sets of 12 figures each, designed to assess fluid intelligence – the CPM was used for children under the age of 11 years and the SPM was used for adults. Each figure is a rectangle filled with geometrical shapes and a piece that is missing in a fixed location. The participants are required to choose between six optional complementary parts, with only one part being the correct answer. The final score is the sum of correct responses (0 = *incorrect*; 1 = *correct*) with maximum score of 36 points. Lastly, mothers' and children's age, as well as children's sex were examined.

### Analytic plan

Participants who did not complete the questionnaires or failed the alertness checks were removed from the sample. First, as a preliminary analysis, a composite factor of cooperativeness was derived from TCI self-reports of CO and the coded behavioral data. This was done in order to have a more ecological and objective measurement of the mothers' cooperativeness. Then, zero-order correlations were examined to choose covariates to control in the main analyses. Then, to examine the main study hypothesis, Hayes's mediation cascade model (Model 6) was used in order to test for mediation beginning with the EC x CO interaction and ending with children's conduct problems, going through mothers' expressive encouragement and children's EC. Then, simple slopes analyses were performed to evaluate the effect of the interaction term. According to a power calculation conducted using G*power, to obtain 95% power, with 5% alpha, two-tailed, with 10 independent variables and a small effect size estimate ($R^2 = .10$), there was a need for $N = 133$ for the analyses to have sufficient power.

### Transparency and openness

All collected data were included in the analysis. The data was collected online via Qualtrics and statistical analysis was done using SPSS 28. All data and the analysis code are available (https://github.com/MaorYeshua/EC-x-CO-ms.git). The study design and analysis were not pre-registered.

## Results

### Preliminary analyses

**Factor analysis.** First, a factor for mothers' CO was formed using all available data ($N = 172$). The factor was loaded by self-reported CO, as measured by the TCI, as well as by the four theory-driven observational codes, which were formulated in light of the definition of CO sub-scales: empathy vs. social disinterest and social acceptance vs. intolerance. Table 2 presents zero-order correlations. As expected, social acceptance and intolerance, as well as empathy and social disinterest, were not mutually exclusive or the absolute reverse of one another.

In line with the theoretical conceptualization, social disinterest and intolerance were positively related ($r = .36$ $p < .001$, $N = 172$). Moreover, and in line with the theoretical construct, the relation between social acceptance and intolerance was negative ($r = −.18$, $p = .018$, $N = 172$), and the relation between empathy and social acceptance was positive ($r = .29$, $p < .001$, $N = 172$). In addition, the relation between social acceptance and intolerance to self-reported CO were found to be significant and in the expected direction ($r = .20$, $p = .009$, $N = 172$; $r = −.18$, $p = .021$, $N = 172$, respectively), while empathy and social disinterest were not ($p$s > .05). The only code that showed counter-intuitive relations to most of the other codes was social disinterest. Empirically, and contrary to the theoretical construct, empathy and social disinterest were positively related ($r = .20$, $p = .010$, $N = 172$).

Factors were extracted using the Principal Component Analysis method, following Varimax rotation with the Kaiser Normalization method. Two factors that exceeded Eigen values of 1 were extracted (see summary in Table 3), while loadings lower than .35 were suppressed to avoid cross-loadings [47,48].

The first factor seemed to better represent CO, as it was better loaded in the expected directions. In line with the inconsistencies of social disinterest with the other scales, it indeed was not loaded into the first factor. The second factor seemed to represent the mothers' uncooperative tendencies, as it was loaded by the mothers' intolerance and social disinterest behaviors.

**Zero-order correlations.** To identify relevant background variables to control for, a zero-order correlations matrix was computed. After searching for a significant relationship with mothers' expressive encouragement, children's EC or conduct problems, the variables that were controlled in the main analysis were household chaos and mothers' Raven score (see Table 4). Moreover, there were high correlations between all six parenting practices ($.15 \leq |r| \leq .62$, $ps < .001$, $N = 168$), and so these were also controlled for in the main analyses.

## Main analysis

**Mediation cascade analysis.** To test the mediation cascade hypothesis, Hayes's Model 6 was used for testing the indirect path between mothers' EC and CO interaction and children's conduct problems. The first mediator was mothers' expressive encouragement, and the second mediator was children's EC; see Fig 1 and Table 5 for the mediation model results.

The analysis revealed that the mothers' EC x CO direct path to children's conduct problems is significant, with and without the mediators. However, the explained variance was greater in the model that included the mediators ($R^2 = .225$) than the model without them ($R^2 = .182$). The mothers' EC x CO interaction significantly explains their expressive encouragement ($p = .010$). In turn, their expressive encouragement explains the children's EC ($p = .012$), which in turn explains the children's conduct problems ($p = .008$). The mediation cascade path was found to be significant ($B = -.01$, $95\%$ Bootstrap CI $[-.025, -.001]$).

## Simple slopes analysis

In order to explore the effect of the interaction between EC x CO on mothers' expressive encouragement and children's conduct problems, two simple slopes analyses were done; see Figs 2 and 3. The first simple slopes analysis revealed that when mothers' CO is low, there is no relationship between mothers' EC and expressive encouragement ($\beta = -.06$, $t(152) = -.68$, $p = .498$, $95\%$ CI $[-.25, .12]$). When mothers' CO is high, the slope of mothers' EC is significant and positive ($\beta = .25$, $t(152) = 3.03$, $p = .003$, $95\%$ CI $[.09, .42]$).

**Table 3.** Cooperativeness factor loadings matrix.

| Variables | Rotated component matrix | |
|---|---|---|
| | **1** | **2** |
| Cooperativeness | .53 | |
| Social acceptance | .76 | |
| Intolerance | −.36 | .78 |
| Empathy | .68 | |
| Social disinterest | | .84 |
| Variance | 29.95% | 27.67% |

$N = 172$. Higher values point to more extreme responses in the expected direction. Cooperativeness was self-reported using the Temperament and Character Inventory (TCI) and all other variables are observation-based scores in the Etch-A-Sketch task.

**Table 4. Zero-order correlation between study variables.**

| | 1. | 2. | 3. | 4. | 5. | 6. | 7. | 8. | 9. | 10. | 11. | 12. | 13. | 14. | 15. | 16. |
|---|---|---|---|---|---|---|---|---|---|---|---|---|---|---|---|---|
| **Background variables** | | | | | | | | | | | | | | | | |
| 1. SES | – | | | | | | | | | | | | | | | |
| 2. CHAOS | .00 | – | | | | | | | | | | | | | | |
| 3. Children's sex | .08 | .06 | – | | | | | | | | | | | | | |
| 4. Children's age | −.08 | −.02 | .04 | – | | | | | | | | | | | | |
| 5. Children's Raven | −.02 | −.07 | −.20** | .38*** | – | | | | | | | | | | | |
| 6. Mothers' age | .06 | .07 | .06 | .01 | −.08 | – | | | | | | | | | | |
| 7. Mothers' Raven | .34*** | .17* | −.13 | −.05 | .21** | −.10 | – | | | | | | | | | |
| **Indexes – Mothers** | | | | | | | | | | | | | | | | |
| 8. EC | .07 | −.39*** | −.17* | .07 | .21** | −.03 | −.06 | – | | | | | | | | |
| 9. CO factor | .23** | −.12 | .00 | −.02 | .05 | .00 | .20** | .16* | – | | | | | | | |
| 10. EE | −.04 | −.20* | −.09 | .09 | .06 | .01 | −.02 | .21** | .16* | – | | | | | | |
| 11. EF | .01 | −.10 | .09 | −.05 | −.03 | .00 | −.16* | .01 | .00 | .40*** | – | | | | | |
| 12. PF | .06 | −.22** | −.10 | −.01 | −.06 | −.02 | −.06 | .12 | .10 | .62*** | .62*** | – | | | | |
| 13. DR | .02 | .27*** | .05 | −.06 | .01 | −.04 | .04 | −.24** | −.17* | −.39*** | −.15 | −.35*** | – | | | |
| 14. PR | −.11 | .21** | .12 | −.09 | .02 | −.02 | −.03 | −.12 | −.23** | −.35*** | −.08 | −.24** | .54*** | – | | |
| 15. MR | .01 | .03 | .11 | −.08 | −.03 | .03 | −.17* | −.05 | −.18* | −.23** | .22** | −.03 | .31*** | .66*** | – | |
| **Indexes – Children** | | | | | | | | | | | | | | | | |
| 16. EC | .02 | −.21** | .10 | −.02 | .02 | −.02 | .02 | .10 | .20** | .30*** | .21** | .26*** | −.15 | −.11 | .08 | – |
| 17. ConP | −.02 | .30*** | −.13 | −.05 | −.04 | −.07 | .11 | −.23** | −.22** | −.13 | −.10 | −.10 | .10 | .05 | −.06 | −.31*** |

$168 \leq N \leq 175$. Higher values point to more extreme responses in the expected direction. Children's sex was a dummy value (0 = *Male*, 1 = *Female*). SES = socioeconomic status factor; EC = effortful control; CHAOS = Confusion, Hubbub, and Order Scale; CO = cooperativeness factor; EE = expressive encouragement; EF = emotion-focused response; PF = problem-focused response; DR = distressed response; PR = punitive response; MR = minimization response; ConP = conduct problems. *$p < .05$, **$p < .01$, ***$p < .001$, two-tailed.

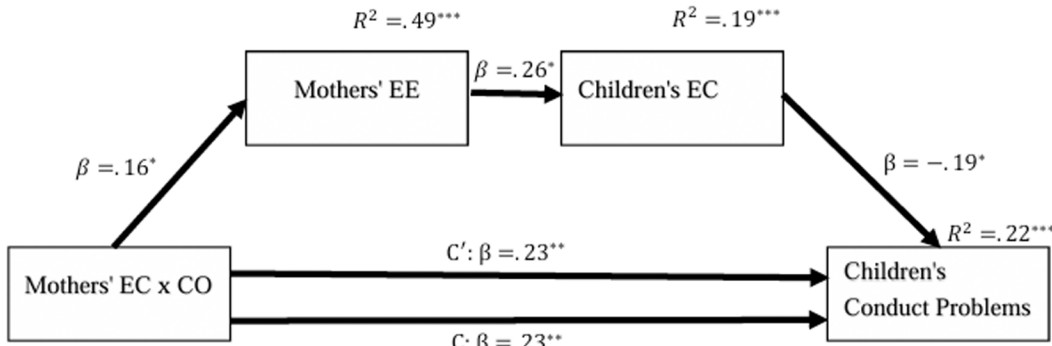

**Fig 1. Mediation cascade model for explaining children's conduct problems.** $N = 163$. Higher values point to more extreme responses in the expected direction. Controlled variables include household chaos, mothers' Raven score, and all five scales of parenting practices. EC = effortful control; CO = cooperativeness; EE = expressive encouragement. *$p < .05$, **$p < .01$, ***$p < .001$, two-tailed.

The second simple slopes analysis revealed that when mothers' CO is low, the slope of mothers' EC is significant ($\beta = −.32$, $t(150) = -2.72$, $p = .008$, 95% CI [-.55, -.09]). When mothers' CO is high, the slope of mothers' EC is not significant and the child's conduct problems are generally below average ($\beta = .02$, $t(150) = .23$, $p = .815$, 95% CI [-.18, .24]).

**Table 5. Mediation cascade analysis.**

| Independent variable | Dependent variable | β | t | p value | 95% CI | | Model summary |
|---|---|---|---|---|---|---|---|
| | | | | | LL | UL | |
| Mothers' EC x CO | Children's conduct problems | .17* | 2.21 | .029 | .02 | .32 | $R^2 = .182$ |
| Mothers' EC | | −.14 | −1.68 | .094 | −.30 | .03 | $F(10, 152) = 3.39$ |
| Mothers' CO | | −.21** | −2.65 | .009 | −.37 | −.05 | $p = .001$ |
| Mothers' EC x CO | Mothers' EE | .16** | 2.62 | .010 | .04 | .28 | $R^2 = .488$ |
| Mothers' EC | | .10 | 1.57 | .120 | −.03 | .23 | $F(10, 152) = 14.49$ |
| Mothers' CO | | .05 | .81 | .418 | −.07 | .17 | $p < .001$ |
| Mothers' EC x CO | Children's EC | −.08 | −.98 | .327 | −.24 | .07 | $R^2 = .185$ |
| Mothers' EC | | −.06 | −.70 | .482 | −.22 | .11 | $F(11, 151) = 3.11$ |
| Mothers' CO | | .12 | 1.53 | .128 | −.04 | .28 | $p = .001$ |
| Mothers' EE | | .27* | 2.55 | .012 | .06 | .47 | |
| Mothers' EC x CO | Children's conduct problems | .17* | 2.20 | .029 | .02 | .33 | $R^2 = .225$ |
| Mothers' EC | | −.14 | −1.72 | .087 | −.30 | .02 | $F(12, 150) = 3.62$ |
| Mothers' CO | | −.18* | −2.28 | .024 | −.34 | −.02 | $p < .001$ |
| Mothers' EE | | −.05 | −.49 | .628 | −.26 | .15 | |
| Children's EC | | −.21** | −2.67 | .008 | −.37 | −.06 | |

*N* = 163. Higher values point to more extreme responses in the expected direction. Controlled variables include household chaos, mothers' Raven score, and all five scales of parenting practices. EC = effortful control; CO = cooperativeness; EE = expressive encouragement. *$p$ < .05, **$p$ < .01, ***$p$ < .001, two-tailed.

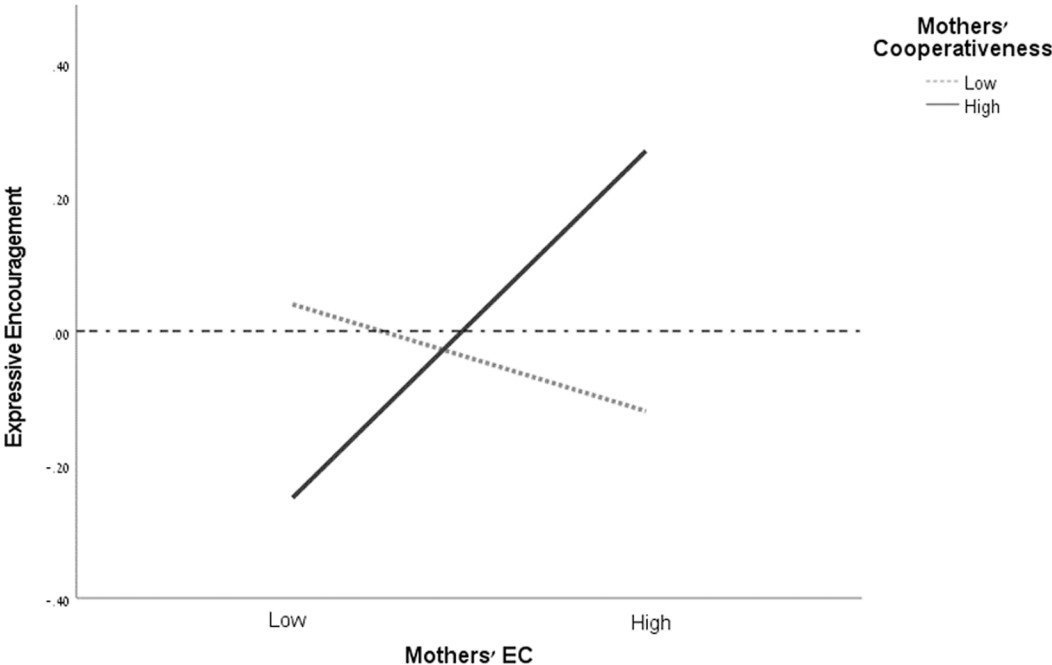

**Fig 2. Mothers' effortful control and cooperativeness interaction explaining mothers' expressive encouragement.** *N* = 163. The horizontal dotted line represents the mean value of the mothers' expressive encouragement. The solid line represents a significant result. EC = effortful control. High = 1 *SD* above the mean of mothers' cooperativeness/effortful control, Low = 1 *SD* below the mean of mothers' cooperativeness/effortful control.

**Fig 3. Mothers' effortful control and cooperativeness interaction explaining children's conduct problems.** $N = 163$. The horizontal dotted line represents the mean value of the child's conduct problems. The solid line represents a significant result. EC = effortful control. High = 1 $SD$ above the mean of mothers' cooperativeness/effortful control, Low = 1 $SD$ below the mean of mothers' cooperativeness/effortful control.

## Discussion

We sought to better understand the complex and multi-layered dynamics in mother-child interactions in self-regulation of emotions and behavior in mother-child dyads performing a task that required cooperation and self-control. Specifically, in kindergarten-aged children (ages 3–6), we tested whether a mother's sensitivity and support in encouraging their children to express their emotions was explained by the interaction of personality measures for cooperation and effortful control of their own emotions and behaviors. In addition, we evaluated whether such controlled and empathic encouragement also contributed to the association of the mother's effortful control and cooperativeness with their children's own effortful control and conduct problems. We found that mother's EC and CO were jointly associated with their encouragement of emotional expression by their children as well as their children's EC and conduct problems.

While previous studies have made significant contributions to our understanding of personality traits and their relation to parent-child dynamics—primarily using self-report measures—the present study offers two key advancements. First, we employed a more ecologically valid and robust method to assess the trait of cooperativeness, utilizing observational data rather than self-report alone. This approach allows for a richer, behaviorally grounded understanding of this personality dimension. Second, our study is among the first to explore in greater depth the interaction between personality traits and how they manifest in both parental and child behaviors. This integrative perspective provides new insights into the dynamic and bidirectional nature of parent-child relationships. The findings revealed that the interaction between mothers' EC and CO partially explained the children's conduct problems, with and without the presence of the mediators. The interaction also partially explained mothers' practice of expressive encouragement, which in turn contributed to the association between the behavior and personality of the mothers and their children. This suggests that expressive encouragement may serve as a resilience factor for enhancing children's EC and reducing conduct problems. Moreover, since this is the first study to offer behavioral measurement for two sub-scales of self-reported CO, as measured by the TCI, it is important

to point out that the sub-scales were found to be reliable and valid, and they converged with the self-reported CO into one factor of CO.

## Cooperativeness behavioral factor

An important innovation of this study is the offering of a behavioral measurement for the sub-scales comprising the CO trait, as defined in the Temperament and Character model [24]. Our exploratory factor analysis was found to be valid and converged with the self-reported CO. This empirical measure combining self-reported and behavioral measurements contributes to the field of character trait measurement via the integration of behavioral observations. This integrative view is in line with Roberts' [49] perspective for personality assessment that suggested personality is composed of thought and feeling, which could be assessed by questionnaires, as well as by behaviors. Furthermore, in line with Cloninger et al.'s [42] view of social acceptance vs. intolerance, our empirical findings indicated that although they oppose one another, in an ecological setting they can be expressed by the same individual in different situations over time, so they are not mutually exclusive. Regarding empathy vs. social disinterest, we found a positive relation between the two, which may be in contradiction to their theoretical definition; thus, the trait vs. state might rely on the frequency of each of these behavioral poles. However, behavioral social disinterest was not loaded into the factor of CO, and it was loaded into a separate factor of uncooperative behaviors. One possible explanation is that the factor that was formulated by social disinterest and intolerance pointed to a separate and unique aspect of uncooperative behaviors.

There is a great need to assess traits in a more comprehensive way that includes reported and behavioral measures, as there are limitations to each level of information to consider. Cooperativeness is a socially desirable characteristic, so many people tend to describe themselves as more cooperative than they really are, as indicated by skewing toward high scores in self-reports and relations with social desirability measures [50]. On the other hand, objective behavioral observations do not give direct awareness to the subjective thoughts and feelings of another person (i.e., to their subjective motives and intentions) [51]. Meaning, relying only on a sole indicator might compromise the external validity of the results.

In light of the fundamental differences in the perspectives of the two methodologies of assessment offered, the weak—however meaningful—correlations that were found in the current study between the reported and the behavioral scales stand out. Besides the methodological differences addressed above, it is also important to note that the reported CO is computed based on five sub-scales, which further justify the nature of correlations found in this study. Lastly, the behavioral measures allowed observation of changes in communication between mother and child that documented the fluctuation in their interactions in a real-life situation.

## Mothers' parenting practices

When mothers behaviorally displayed lower levels of CO, there was no relation between their EC and expressive encouragement. However, when they displayed high levels of CO and had high levels of EC, the mothers demonstrated the highest levels of expressive encouragement of their children. This finding supported the theoretical models that were suggested by Yeshua and Berger [21], as well as by Edler and Valentino [27]. Mothers who are characterized with high tendency to be empathic, tolerant and compassionate toward their child [24,52] have parenting practices that are more adaptive, as their ability to exert inhibitory control and executive attention is higher (i.e., higher self-regulation; [7,12]). It is likely that when such mothers experience an aversive situation with their child, they are more resilient in coping with stress calmly or in adapting to the negative emotions the situation may arouse in them and/or in their child (better EC) and more empathic and tolerant (as they are high on CO) [53,54]. As a result, they can actively encourage their child to express negative emotions [25] because they can reassure and identify with their child. This theoretical notion is also in line with Scott and Hakim-Larson's [26] research showing that lower rejection and disapproval of a child's emotions is identified in mothers who show better EC and lower negative affectivity.

The literature supports partial contributions of EC and CO on parenting practices and styles [9,28–30]. Our findings indicate that the effect of EC depends on its interaction with CO, and that their joint effect on their children's behavior further depends on the development of caregiving abilities such as expressive encouragement. Yet, it is important to note that our findings are based on a specific type of parenting practice, which has not been specifically tested in the mentioned studies.

## Children's conduct problems

Our mediation cascade model suggests that the interaction between mothers' EC and CO explained their expressive encouragement, which in turn explained the children's EC and conduct problems. This multi-step path elaborates on the existing literature that demonstrated the mediating effects of parenting on children's effortful control and conduct problems [9,10,29], as well as on emotion regulation and emotional lability [26,31]. It is also in line with Bridgett et al.'s [8] model regarding the intergenerational transmission of self-regulation, as well as with the findings in the more general field of maternal sensitivity [32–34]. It is in accord with findings of Cloninger and his colleagues about the self-regulation of genetic-environmental interactions in the epigenetic development of temperament and character, which involves multiple systems of learning that have evolved in a stepwise manner [24,52,55,56]. These findings indicate the general notion that the different personality components of temperament and character are valuable for explaining parenting practices and help in shaping the intergenerational transmission of effortful control. They also emphasize the idea of the caregivers' sensitive parenting being a buffer against aversive behavior in children [57]. Not only is the interaction being mediated by parenting and children's effortful control, but it also has a direct effect on children's conduct problems.

When mothers were found to be high on CO, there was no relation between mothers' EC and their children's conduct problems, and the average level of children's conduct problems was below the mean. On the other hand, when mothers were found to be low on CO, there was a relation between mothers' EC and their children's conduct problems—when the mothers were also rated low on EC, their children were rated high on conduct problems. This finding implies that the moderating effect of empathy, social acceptance and compassion on inhibition and executive attention also explains lower levels of "acting out" behaviors (i.e., conduct problems; [58]). The literature suggests that the mothers' personality is not directly related to the child's behavior, rather it goes through, to some extent, their parenting practices and style [8–10,21]. Hence, it is possible that this direct path can be explained by other parenting practices that were not measured in the current study, and is being mediated by them, affecting children's effortful control and conduct problems.

## Applied implications

Although the present design and results do not permit causal conclusions regarding the influence of maternal EC and CO on parenting behavior as well as child behavior, it is possible to speculate on such suggestions. The recognition of elements within the child's microsystem, as well as the pathways through which mothers contribute to shaping this environment and thereby affect their children, lends support to the rationale behind mother-centered interventions. More specifically, these findings underscore the inherent complexity of providing parental guidance within therapeutic settings. When considering that parental sensitivity is possibly rooted in meta-stable maternal personality traits (e.g., EC and CO), such insights offer a valuable framework for refining intervention strategies. Such personality traits are not fixed traits as has sometimes been falsely assumed; rather they are meta-stable, which means that they change in a step-wise manner under appropriate conditions that facilitate change [24]. Understanding the necessary conditions for change will benefit from through consideration of the multi-step nature of personality development in children. For instance, recognizing a parent's difficulties with inhibition can lead to the introduction of structured tools aimed at improving executive functioning in daily parenting tasks. Similarly, identifying maternal challenges related to empathy or cooperative tendencies opens the door to interventions such as reflective techniques that help the parent attune to the child's internal states, thereby fostering greater empathy and responsiveness. These considerations are particularly salient during the preschool years—the

developmental stage examined in the current study—which have been widely recognized as a critical period for shaping long-term developmental trajectories. The stability of the parent-child dyadic relationship, extending from early childhood into adulthood [59], further underscores the significance of early, tailored, and personality-informed parental guidance.

### Limitations and future studies

The present study has several limitations that should be noted. First, the sample is based on mothers only. Hence, our findings cannot be generalized to fathers, as there are findings supporting differences in the relation between personality dimensions and parenting between the genders [60,61]. Regarding the factor of cooperativeness and the new behavioral coding, there are limits to what either overt behavior or self-reports can reveal when used in isolation. Consider the rated behaviors that suggested possible "social disinterest," such as "Why are you so frustrated? It is only a game." The same statement may express either disinterest or reassurance. For example, such a statement can be made by a cooperative mother being respectful and empathic in letting go of control of her child in a safe, playful situation. Alternatively, a mother who likes to think of herself as empathic may become frustrated and interact with her child in an unhelpful or disinterested way. Hence, future research can be refined by attention to tone of voice and facial micro-expressions in order to detect changes in unconscious expressed emotions; but these are ultimately inferences that have limitations. Moreover, further research is needed to clarify the limitations of behavioral indices of subjective motivation, as well as to assess the value of combining self-report measures with objective personality indices because the combined use can be valuable in both research and clinical practice. Since the observational variables estimating CO were formulated and analyzed here for the first time, their validity should be replicated and their limitations in capturing subjective motivation should be further explored. A longitudinal design would allow examination of the stability of the behavioral coding and its relation to self-reported cooperativeness. In addition, it would be informative to conduct the behavioral coding in diverse participant populations.

Moreover, the current study focused on a specific interaction of temperament and character in the forms of EC and CO. The concept of temperament and character should be further studied, looking for additional moderations between other traits. Furthermore, the findings concern mothers of preschool- and kindergarten-aged children. Hence, these hypotheses should be tested also on parents and their older-aged children and adolescents. Lastly, although the study's theoretical model is based on previous well-established theoretical models and empirical findings, our cross-sectional design limits the ability to draw developmental and temporal inferences from the mediation analysis performed.

## Conclusions

Mothers' effortful control interacts with their cooperativeness when explaining their parental practices, which, in turn, partly explains their children's effortful control and conduct problems. This is the first study offering a behavioral assessment of cooperativeness and the first to test its moderation effects within the area of parenthood. The results suggest that these different aspects of personality interact in ways that explain mothers' behavior, which acts as a buffer against aversive behavior of their children. Our finding help to clarify the complex and multi-step ways by which a mother's temperament and character may influence the development of personality and psychopathology in her children.

## Supporting information

**S1 Appendix. Etch-A-Sketch – Cooperativeness codes description.**
(DOCX)

## Acknowledgments

We would like to thank Mrs. Desiree Meloul for the review and formatting of the manuscript.

## Author contributions

**Conceptualization:** Maor Yeshua, Claude Robert Cloninger, Ada H. Zohar, Andrea Berger.

**Data curation:** Maor Yeshua.

**Formal analysis:** Maor Yeshua.

**Funding acquisition:** Andrea Berger.

**Investigation:** Maor Yeshua.

**Methodology:** Maor Yeshua.

**Project administration:** Maor Yeshua.

**Resources:** Andrea Berger.

**Software:** Maor Yeshua, Andrea Berger.

**Supervision:** Andrea Berger.

**Visualization:** Maor Yeshua.

**Writing – original draft:** Maor Yeshua.

**Writing – review & editing:** Maor Yeshua, Claude Robert Cloninger, Ada H. Zohar, Andrea Berger.

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
