## [Decision Letter · Decision Letter 0]

7 May 2025

PONE-D-25-08839Extending the Model of Children's Conduct Problems: A Cross-sectional Study of the Interaction of Maternal Temperament and Character, Maternal Parenting Practices, and Their Child's Effortful ControlPLOS ONE

Dear Dr. Yeshua,

Thank you for submitting your manuscript to PLOS ONE. After careful consideration, we feel that it has merit but does not fully meet PLOS ONE’s publication criteria as it currently stands. Therefore, we invite you to submit a revised version of the manuscript that addresses the points raised during the review process.

Besides the Reviewers comments (INCLUDED BELOW), I would like that readers may be more informed about the meaning of the methodological and analytical rational. On the one hand, the authors assess Adult Temperament using Rothbart's model, which includes a dimension of effortful control (wiich turns out to play a central role in this study). On the other hand, the authors also evaluate the TCI Character dimensions of Self-directedness and Cooperativeness. Then, the author conduct different analyses including (mediation analyses) amongst effortful control and these two character dimensions. For what we know about the dimension of Effortful control refers to a dimension that involves intentional (not only reactive dispositional tendencies = temperament) higher order self-refulatory processes, which overlap with the dimensions of self-directedness and cooperativeness.

I think that both the rational of the study and the results would benefits from authors to elaborate on the problematization about the conceptual/contructrational and meaning of this question, reason why I invite the authors to do it.

We look forward to receiving your revised manuscript.

Kind regards,

Paulo Alexandre Soares Moreira, PhD

Academic Editor

PLOS ONE

3.** ** Thank you for stating the following financial disclosure:

 [This study was partially supported by the Israel Scientific Foundation (ISF; Grant number 533/20).]. 

5. Please ensure that you include a title page within your main document. You should list all authors and all affiliations as per our author instructions and clearly indicate the corresponding author.

Reviewers' comments:

Reviewer's Responses to Questions

**Comments to the Author**

1. Is the manuscript technically sound, and do the data support the conclusions?

Reviewer #1: Partly

Reviewer #2: Yes

2. Has the statistical analysis been performed appropriately and rigorously? 

Reviewer #1: I Don't Know

Reviewer #2: Yes

3. Have the authors made all data underlying the findings in their manuscript fully available?

Reviewer #1: Yes

Reviewer #2: Yes

4. Is the manuscript presented in an intelligible fashion and written in standard English?

Reviewer #1: Yes

Reviewer #2: Yes

5. Review Comments to the Author

Reviewer #1: Abstract

The abstract is engaging in topic but does not adhere to the structured IMRD (Introduction, Methods, Results, Discussion) format, which is a critical requirement for understanding scientific articles.

Key methodological details such as measurements and statistical analyses are missing or hard to find and it made this article difficult for readers to quickly grasp the study's design.

The results section in the abstract is vague; specific findings, including statistical significance and effect sizes, should be briefly mentioned.

The importance of the Etch-A-Sketch observational coding is noted, yet its introduction in the abstract or introduction section feels abrupt and lacks sufficient context.

The keywords are appropriate, but adding terms related to "maternal character" and "Etch-A-Sketch observation" might enhance discoverability.

Introduction

The introduction presents an important research topic but suffers from excessive citation of previous studies without offering a clear, succinct synthesis.

Definitions of effortful control (EC) and cooperativeness (CO) are individually provided but are not sufficiently integrated into a coherent conceptual framework.

Several sections seem to directly reproduce the contents of prior research articles rather than organizing them in a consistently structured narrative.

The number of theoretical concepts introduced is overwhelming; many may not be essential for this study and should be streamlined.

If all introduced concepts are indeed necessary, organizing them in a table or diagram would enhance clarity and reader comprehension.

The rationale for using the Etch-A-Sketch as a behavioral measure of CO is meaningful but is introduced too late; it should be presented earlier in the Introduction.

The novelty of integrating behavioral and self-reported measures is noteworthy but could be articulated more succinctly to highlight the study's contribution.

Method

The method section lacks precision in the description of statistical analyses. Readers are left unclear about which specific methods were applied to which variables.

Descriptions of measurement tools are lengthy and fragmented. A succinct table summarizing each tool, its subscales, and rationale for using would greatly aid understanding.

The reason for employing Principal Component Analysis (PCA) to derive the CO factor is not sufficiently explained and should be justified more clearly.

The order of explaining analytic methods (mediation cascade vs. factor analysis) is confusing. Aligning the sequence logically is recommended.

The selection of control variables is not fully transparent. Clearer justification based on theoretical or empirical grounds would strengthen this section.

Measures of the Etch-A-Sketch task are creative but overly detailed. A summarizing figure or table would make it easier to follow.

Ethical procedures are appropriately reported but would benefit from a more concise presentation.

Sample characteristics are provided, but comparisons with general population demographics would help in evaluating external validity.

Results

The results section lacks clear subheadings (e.g., "Factor Analysis," "Mediation Analysis," "Simple Slopes Analysis") that would aid in structuring the presentation.

Descriptions of findings are often embedded in lengthy paragraphs. Breaking into smaller sections with descriptive headings would improve readability.

Although mediation analysis is conducted, the presentation of coefficients and confidence intervals could be more systematically organized in a table.

The description of simple slope analyses is technically correct but would benefit from graphical enhancements with higher resolution figures, as the current images are too blurry to interpret properly.

The sequence of presenting results does not fully align with the hypotheses stated earlier. A more coherent flow would strengthen the narrative.

Discussion

The discussion offers insightful interpretations but at times reiterates the introduction rather than critically evaluating the findings.

Lines 400-411 discussing limitations should be moved into a clearly labeled “Limitations” section toward the end of the discussion.

The use of "temperament and character" throughout the paper is somewhat inconsistent. Since only CO (a character trait) was behaviorally assessed, it would be more accurate to emphasize "character."

While the study’s contribution in developing a behavioral measure for cooperativeness is valuable, the discussion would benefit from a more balanced reflection on the limitations of behavioral observation.

Future directions are mentioned but could be expanded to suggest specific improvements to the methodology, including longitudinal designs and more diverse participant populations.

Reviewer #2: This research paper is interesting enough to be a source of reading in journal publications. However, there are several things that need to be considered to be improved so that this paper becomes better. First, the writing of the paper should be written using the concept of "right alignment" in each paragraph that is displayed. Second, the writing of numbers from reference sources should be written above the last word of each closing sentence per paragraph. Third, the author should display one additional paragraph explaining the theoretical review based on literature sources which are the main principles of the relationship between the factors studied in the study. Fourth, the author needs to add in the discussion the advantages of this research when compared to other or previous studies. Fifth, the author needs to make corrections again regarding the writing of the order of reference materials contained in the literature sources. In addition, researchers can write a little about the contribution of the results of this study to the development of awareness programs and prevention of conduct problems in children and adolescents in the future.

6. PLOS authors have the option to publish the peer review history of their article (what does this mean? ). If published, this will include your full peer review and any attached files.

**Do you want your identity to be public for this peer review?** For information about this choice, including consent withdrawal, please see our Privacy Policy .

Reviewer #1: No

Reviewer #2: **Yes: ** Isa Multazam Noor, M.Sc., M.D., (Child Psych)

---

## [Author Response · Author response to Decision Letter 1]

18 Jun 2025

Dear Editor,

We thank you and the reviewers for your important and valuable comments on our manuscript. We have addressed the comments in bold text after each issue that was raised.

1. Please ensure that your manuscript meets PLOS ONE's style requirements, including those for file naming. The PLOS ONE style templates can be found at https://journals.plos.org/plosone/s/file?id=wjVg/PLOSOne_formatting_sample_main_body.pdf and https://journals.plos.org/plosone/s/file?id=ba62/PLOSOne_formatting_sample_title_authors_affiliations.pdf.

Thank you for drawing our attention to this issue. We have verified the requested requirements and applied changes as necessary.

Thank you that remark; we have made sure the funders are spelled identically, in lines 142-143:

"the Israel Science Foundation (grant number 533/20)."

[This study was partially supported by the Israel Scientific Foundation (ISF; Grant number 533/20).].

We have added the required statement to the cover letter.

Thank you for that note. We have made the full data available on GitHub (https://github.com/MaorYeshua/EC-x-CO-ms.git). The link now appears in the full text of the manuscript. Under the "Additional data availability information:" we have removed the first tick, as the data in now available.

5. Please ensure that you include a title page within your main document. You should list all authors and all affiliations as per our author instructions and clearly indicate the corresponding author.

We have included a title page within the main document.

6. Besides the Reviewers comments (INCLUDED BELOW), I would like that readers may be more informed about the meaning of the methodological and analytical rational. On the one hand, the authors assess Adult Temperament using Rothbart's model, which includes a dimension of effortful control (which turns out to play a central role in this study). On the other hand, the authors also evaluate the TCI Character dimensions of Self-directedness and Cooperativeness. Then, the author conduct different analyses including (mediation analyses) amongst effortful control and these two character dimensions. For what we know about the dimension of Effortful control refers to a dimension that involves intentional (not only reactive dispositional tendencies = temperament) higher order self-regulatory processes, which overlap with the dimensions of self-directedness and cooperativeness.

I think that both the rational of the study and the results would benefits from authors to elaborate on the problematization about the conceptual/contructrational and meaning of this question, reason why I invite the authors to do it.

Thank you for this valuable and important comment. We acknowledge the overlap between effortful control and the dimensions of self-directedness and cooperativeness, and specifically with self-directedness. As self-directedness is not the focus of this study, we have expanded our rational regarding the relationship between effortful-control and cooperativeness specifically (p. 5 lines 67-90).

Reviewer #1:

1. Abstract

The abstract is engaging in topic but does not adhere to the structured IMRD (Introduction, Methods, Results, Discussion) format, which is a critical requirement for understanding scientific articles.

Key methodological details such as measurements and statistical analyses are missing or hard to find and it made this article difficult for readers to quickly grasp the study's design.

The results section in the abstract is vague; specific findings, including statistical significance and effect sizes, should be briefly mentioned.

The importance of the Etch-A-Sketch observational coding is noted, yet its introduction in the abstract or introduction section feels abrupt and lacks sufficient context.

The keywords are appropriate, but adding terms related to "maternal character" and "Etch-A-Sketch observation" might enhance discoverability.

We thank the reviewer for the valuable comment. We have added headings in order to follow the structured IMRD format. Moreover, we have added the requested information to the methods and results sections. We have also added the recommended keywords.

2. Introduction

The introduction presents an important research topic but suffers from excessive citation of previous studies without offering a clear, succinct synthesis.

Definitions of effortful control (EC) and cooperativeness (CO) are individually provided but are not sufficiently integrated into a coherent conceptual framework.

Several sections seem to directly reproduce the contents of prior research articles rather than organizing them in a consistently structured narrative.

The number of theoretical concepts introduced is overwhelming; many may not be essential for this study and should be streamlined.

If all introduced concepts are indeed necessary, organizing them in a table or diagram would enhance clarity and reader comprehension.

The rationale for using the Etch-A-Sketch as a behavioral measure of CO is meaningful but is introduced too late; it should be presented earlier in the Introduction.

The novelty of integrating behavioral and self-reported measures is noteworthy but could be articulated more succinctly to highlight the study's contribution.

We would like to thank the reviewer for these important comments. As a result, we have rearranged and fully edited the Introduction section.

3. Method

The method section lacks precision in the description of statistical analyses. Readers are left unclear about which specific methods were applied to which variables.

Descriptions of measurement tools are lengthy and fragmented. A succinct table summarizing each tool, its subscales, and rationale for using would greatly aid understanding.

The reason for employing Principal Component Analysis (PCA) to derive the CO factor is not sufficiently explained and should be justified more clearly.

The order of explaining analytic methods (mediation cascade vs. factor analysis) is confusing. Aligning the sequence logically is recommended.

The selection of control variables is not fully transparent. Clearer justification based on theoretical or empirical grounds would strengthen this section.

Measures of the Etch-A-Sketch task are creative but overly detailed. A summarizing figure or table would make it easier to follow.

Ethical procedures are appropriately reported but would benefit from a more concise presentation.

Sample characteristics are provided, but comparisons with general population demographics would help in evaluating external validity.

Thank you for the important comments. First, we have added, in the Sample characteristics, comparisons with the general population demographics.

"The mothers' ages ranged between 28-51 years (M = 36.69 ± 4.45) and years of education ranged between 7-30 years (M = 17.62 ± 3.16), which when compared to national data, indicates the mothers in the current sample were slightly more educated [35]. Their household income was diverse with half having higher/lower income compared to the monthly demographic average…"

We have also added a table (Table 1) that summarizes the variables that were included in the model and their role in the main analysis. Regarding the Cooperativeness codes in the Etch-A-Sketch task, we have shortened the section and referenced an appendix in order to make it easier to follow in the main text. As for the justification for controlled variables, we have added references in order to better justify their selection as possible covariates. In addition, we have added justifications for selecting the possible covariates described in the background variables section. Regarding the ethical procedure, we have revised it to be more concise and yet follow the journal requirements.

"As socioeconomic status in a risk factor for the development of psychiatric disorders [43], we examined it as a possible covariate. … Moreover, as household chaos and intelligence (of mother and child alike) are related to one another and have an effect on the child's outcomes [44, 45], they were also considered as possible covariates…"

Moreover, we have better stated the factor analysis and zero-order correlations as preliminary analyses and the mediation cascade as the main analysis in the analytical plan section. We have also explained the reasoning for creating a factor for cooperativeness.

"First, as preliminary analyses, a composite factor of cooperativeness was derived from TCI self-reports of CO and the coded behavioral data. This was done in order to have a more ecological and objective measurement of the mothers' cooperativeness. … Then, to examine the main study hypothesis, Hayes's mediation cascade model (Model 6) was used …"

4. Results

The results section lacks clear subheadings (e.g., "Factor Analysis," "Mediation Analysis," "Simple Slopes Analysis") that would aid in structuring the presentation.

Descriptions of findings are often embedded in lengthy paragraphs. Breaking into smaller sections with descriptive headings would improve readability.

Although mediation analysis is conducted, the presentation of coefficients and confidence intervals could be more systematically organized in a table.

The description of simple slope analyses is technically correct but would benefit from graphical enhancements with higher resolution figures, as the current images are too blurry to interpret properly.

The sequence of presenting results does not fully align with the hypotheses stated earlier. A more coherent flow would strengthen the narrative.

We appreciate these comments; we have added subheadings as requested and have divided the paragraphs into shorter parts. Moreover, we have regenerated the graphs with a higher resolution. We have also aligned the text with the hypothesis section and analytical plan. In addition, we have shortened the statistical report in the text on the mediation cascade and added a summarizing table.

5. Discussion

The discussion offers insightful interpretations but at times reiterates the introduction rather than critically evaluating the findings.

Lines 400-411 discussing limitations should be moved into a clearly labeled “Limitations” section toward the end of the discussion.

The use of "temperament and character" throughout the paper is somewhat inconsistent. Since only CO (a character trait) was behaviorally assessed, it would be more accurate to emphasize "character."

While the study’s contribution in developing a behavioral measure for cooperativeness is valuable, the discussion would benefit from a more balanced reflection on the limitations of behavioral observation.

Future directions are mentioned but could be expanded to suggest specific improvements to the methodology, including longitudinal designs and more diverse participant populations.

Thank you for the important comment; we have moved lines 400-411 to the limitation section. In addition, we have added the following to the limitations section:

" … Hence, future research can be refined by attention to tone of voice and facial micro-expressions in order to detect changes in unconscious expressed emotions; but these are ultimately inferences that have limitations. Moreover, further research is needed to clarify the limitations of behavioral indices of subjective motivation, as well as to assess the value of combining self-report measures with objective personality indices because the combined use can be valuable in both research and clinical practice. Since the observational variables estimating CO were formulated and analyzed here for the first time, their validity should be replicated and their limitations in capturing subjective motivation should be further explored. A longitudinal design would allow examination of the stability of the behavioral coding and its relation to self-reported cooperativeness. In addition, it would be informative to conduct the behavioral coding in diverse participant populations."

Moreover, in the Introduction and Discussion we have further clarified that the specific temperamental trait we assessed was EC and that the character trait was CO.

Reviewer #2:

This research paper is interesting enough to be a source of reading in journal publications. However, there are several things that need to be considered to be improved so that this paper becomes better.

1. First, the writing of the paper should be written using the concept of "right alignment" in each paragraph that is displayed.

2. Second, the writing of numbers from reference sources should be written above the last word of each closing sentence per paragraph.

Thank you for your comments; we have further examined the journal guidelines and made sure we have followed their instructions for "right alignment" and within-text referencing.

3. Third, the author should display one additional paragraph explaining the theoretical review based on literature sources which are the main principles of the relationship between the factors studied in the study.

Thank you for this comment; we have revised the Introduction section to better describe the theoretical basis of this study.

4. Fourth, the author needs to add in the discussion the advantages of this research when compared to other or previous studies.

Thank you for this valuable comment; we have added the following text in the beginning of the Discussion section:

"… While previous studies have made significant contributions to our understanding of personality traits and their relation to parent-child dynamics—primarily using self-report measures—the present study offers two key advancements. First, we employed a more ecologically valid and robust method to assess the trait of cooperativeness, utilizing observational data rather than self-report alone. This approach allows for a richer, behaviorally grounded understanding of this personality dimension. Second, our study is among the first to explore in greater depth the interaction between personality traits and how they manifest in both parental and child behaviors. This integrative perspective provides new insights into the dynamic and bidirectional nature of parent-child relationships …"

5. Fifth, the author needs to make corrections again regarding the writing of the order of reference materials contained in the literature sources.

We have revised the reference list and made sure it follows correct citation rules.

6. In addition, researchers can write a little about the contribution of the results of this study to the development of awareness programs and prevention of conduct problems in children and adolescents in the future.

Thank you for thi

---

## [Editor Report · Decision Letter 1]

8 Aug 2025

Extending the model of children's conduct problems: A cross-sectional study of the interaction of maternal temperament and character, maternal parenting practices, and their child's effortful control

PONE-D-25-08839R1

Dear Dr. Yeshua,

We’re pleased to inform you that your manuscript has been judged scientifically suitable for publication and will be formally accepted for publication once it meets all outstanding technical requirements.

Kind regards,

Paulo Alexandre Soares Moreira, PhD

Academic Editor

PLOS ONE
---

## [Editor Report · Acceptance letter]

PONE-D-25-08839R1

PLOS ONE

Dear Dr. Yeshua,

I'm pleased to inform you that your manuscript has been deemed suitable for publication in PLOS ONE. Congratulations! Your manuscript is now being handed over to our production team.

Kind regards,

on behalf of

Professor Paulo Alexandre Soares Moreira

Academic Editor

PLOS ONE